# A Technological Scenario for a Healthier, More Equitable and Sustainable Europe in 2040: Citizen Perceptions and Policy Implications

**DOI:** 10.3390/ijerph17010231

**Published:** 2019-12-28

**Authors:** Arlind Xhelili, Rosa Strube, Francesca Grossi, Iva Zvěřinová, Timothy Taylor, Pablo Martinez-Juarez, Sonia Quiroga, Cristina Suárez, Dragan Gjorgjev

**Affiliations:** 1Collaborating Centre on Sustainable Consumption and Production (CSCP), 42107 Wuppertal, Germany; rosa.strube@scp-centre.org (R.S.); francesca.grossi@scp-centre.org (F.G.); 2Charles University, Environment Centre (CUNI), 162 00 Prague, Czech Republic; iva.zverinova@czp.cuni.cz; 3European Centre on Environment and Human Health, University of Exeter Medical School, Truro TR1 3HD, UK; timothy.j.taylor@exeter.ac.uk; 4Health Economics Group, University of Exeter Medical School, Exeter EX1 2LU, UK; P.Martinez-Juarez@exeter.ac.uk; 5Department of Economics, Universidad de Alcalá, 28801 Alcalá, Spain; sonia.quiroga@uah.es (S.Q.); cristina.suarez@uah.es (C.S.); 6The Institute of Public Health of the Republic of Macedonia (IJZRM), 1000 Skopje, North Macedonia; dgjorgjev@gmail.com

**Keywords:** sustainability, health, equity, lifestyles, future scenarios, technology, citizen insights, policy recommendations

## Abstract

This article aims at exploring, understanding and comparing European citizens’ insights and perceptions towards “My life between realities”, a positive future scenario which depicts a narrative of reaching healthier, more equitable and sustainable societies by 2040 with the support of technology and technological solutions. It responds to the need for gathering and incorporating more citizen insights into future policy developments and strategic actions to tackle the global challenge of unsustainable development. Citizens of five European countries—the Czech Republic, Germany, North Macedonia, Spain and the United Kingdom—have been consulted through focus groups. The exercise has uncovered citizens’ preferences and attitudes towards four main lifestyle areas; namely, green spaces, energy efficient housing, active mobility and (food) consumption. The technological attributes of the scenario led to citizens expressing diametrically opposed and critical perceptions and attitudes. Given the prospects of technology in driving sustainable development, based on these insights, policy recommendations for the better integration and acceptance of technological advances by the public are discussed herein.

## 1. Introduction

### 1.1. Background on European Development 

Throughout many years of development, and especially the last four decades, humankind, at an immense pace, has continuously and effectively worked towards improving standards and quality of living and well-being, ensuring further progress and testing the limits of development [1]. Accelerated by economic liberalization, industrialization, technological advances and information flow, Western societies have experienced outstanding economic, social, urban and cultural developments [2,3]. Additionally, these advances, cumulatively, have led to changes in consumption and production patterns that have enabled citizens to enjoy a temporally and spatially sustained access to good quality and affordable goods and services [4]. Nonetheless, despite achieving the intended progress, these developments have also led to harmful socioeconomic and environmental impacts and risks, giving rise to inequalities and indicating that our development trajectory is not sustainable in the long-term [3,5,6]. 

Natural resource and material depletion, air, water and soil pollution and greenhouse gas emissions are some of the environmental by-products of our development and production and consumption patterns [3,7]. These patterns, for many decades now, have not been in line with the planetary capacities necessary to support them [8]. In 2015, the global extraction of resources and materials reached 84 billion tons, more than triple the amount of four decades earlier [9]. In 2014, the ecological footprint of the EU-28 was 4.7 global hectares per person compared to a biocapacity of 2.2 global hectares per person, indicating an ecological deficit of −2.5 global hectares per person [10]. Taking a global average biocapacity of 1.7 global hectares per person [11], if all people in the world were to live similar lifestyles with similar demands as the average EU-28 citizen, then just under 3 Earths would be needed to support these depletive patterns. 

Nowadays, as a result of improved living and well-being standards, Europeans enjoy greater health and longer longevity compared to previous years [12]. For example, between 1990 and 2014, the life expectancy in the EU increased from 74.2 to 80.9 years [13]. Nonetheless, at the same time, Europe is facing increasing rates of chronic diseases and multi-morbidity, along with an ageing population. Over the last decade, infectious diseases have been substituted by non-communicable ones, including cardiovascular and respiratory diseases, cancer, diabetes, obesity and mental disorders, and these have become the number one cause of death in the EU [12]. Coupling this with that fact that approximately 19% of Europe’s population is 65 years and over [14], and this proportion is expected to increase further [15], we can see there are significant challenges for the health sector and the wider society. 

Prominent social challenges in the EU and Europe in general are the perpetual inequality levels, including the unequal distributions of income and wealth [16]. Inequality has been shown to have an impact on citizens’ health, and environmental inequalities may be a significant contributing factor, particularly affecting vulnerable social groups [17,18,19]. Inequalities can be observed among and within the EU Members States and are related to citizens’ living circumstances (housing), education, profession (employment), social and financial services and lifestyle quality. For example, in 2018, 21.7% of the EU-28 population were at risk of falling under the poverty line and excluded from the aforementioned opportunities [20]. Air pollution, besides its negative impacts on the environment, negatively influences human health. It has been estimated that, by 2060, air pollution could be the cause of 6 to 9 million premature deaths globally [12], disproportionately affecting vulnerable groups, such as low-income group citizens and the elderly, due to their inability to fully engage in mitigation and adaptation measures [18,19]. 

To decouple these occurrences from development efforts and ensure a just and egalitarian society for both current and future generations, the sustainable development agenda was introduced in the socioeconomic and political discourse [3,5,21]. Since its introduction, several milestones have marked its progress, with the latest being the Sustainable Development Goals (SDGs) that guide our development pathway to 2030 [22,23,24,25]. As such, integrating sustainability in our development patterns has been the subject of various initiatives on different levels and in different areas with varying degrees of effectiveness. In many instances, technological measures are considered to be part of the solution with the greatest potential for effectiveness in reaching this integration. For example, large scale diffusion of renewable energy and energy efficiency are key for the decarbonization of the energy system as an unnegotiable pre-condition to meet the 1.5 °C Paris temperature goal [26]. 

### 1.2. Technological (Digital) Solutions for Sustainable Development 

Starting in early 1950s and advancing gradually throughout the years, the digital revolution has evolved exponentially to include artificial intelligence, virtual and augmented reality, big data, robotics, additive manufacturing, autonomous systems and the Internet of Things [27]. These advances, among others, have an enormous potential to positively contribute towards increased resource efficiency, decarbonization of lifestyle areas (energy, mobility, food) and related industries and the dematerialization of production [27,28]. Additionally, digital technology can help facilitate non-ownership based models of access to services, paving the way for more circular and sharing business models, while supporting our improved understanding of the impact our decisions have throughout various systems and predicting human behavior and related demands, leading to more sustainable behaviors [18,28]. Technological (digital) measures are already disrupting the medical and public health sector. Accordingly, these have the potential to support the increased adoption of healthy lifestyles among the general public by enabling an active monitoring of personal health and providing personalized suggestions for maintaining and/or improving one’s health-related activities [13,19,29,30]. Furthermore, technological solutions and digitalization can support an increased access to care for all citizens and reduce citizens’ overall time spent in health systems, leading to more optimized systems and cost savings [13,29]. The positive aspects of digital and technological measures on the economy are recognized by 75% of respondents of a 2017 Eurobarometer poll, while 67% and 64% believe these measures can positively influence their quality of life and society in general, respectively [31]. Similarly, positive aspects of robots and artificial intelligence are reflected by Europeans’ views, as 68% of respondents of the same Eurobarometer poll [31] (page 65) agreed with the statement that “robots and artificial intelligence are a good thing for society because they help people do their jobs or carry out daily tasks at home.”

Nonetheless, several challenges have been voiced, by citizens as well, in regard to these developments and technology’s increasing influence on our living patterns, potentially limiting technology’s potential for supporting sustainable development. One widespread concern relates to automation’s impact on human jobs and the human workforce [32]. In a Eurobarometer poll [31] (page 6), most Europeans agreed that “more jobs will disappear than new jobs will be created” because of robots and artificial intelligence (74%), and that “robots and artificial intelligence steal people’s jobs” (72%). Moreover, the current technological solutions, due to their high prices and literacy required to operate, might not be affordable, accessible or utilized by all citizens regardless of their socioeconomic and educational status [19]. Europeans with the lowest education levels and with the most difficulties paying bills are more likely to be worried about the loss of jobs due to robots and artificial intelligence [31]. This trajectory can lead to furthering the current socioeconomic divide [19]. Partly due to these phenomena, several measures targeting sustainability have faced severe backlash. Such is the case for carbon taxes targeting conventional fuels. Furthermore, if this technological transformation does not evolve in alignment with sustainability principles, it might accelerate inefficient resource and material use. To offer personalized solutions and increase user experience, technological products and services rely on data collection and analysis. These processes have led to concerns related to violation of human privacy and autonomy [30,32]. The latter are reinforced by the prevalent ambiguity about the collectors of these data, (data) management practices, overseeing bodies and sanction mechanisms in case of empirical misuse [30]. 

Accordingly, the digital revolution, besides holding the disruptive potential of change, has, undoubtedly, challenged all forms of governance [27,33]. These technological developments have irreversibly changed the way humans live, move and consume. In view of these developments and the strong presence of technology in the present and the foreseen growth in the future, it is of great importance that the future development efforts integrate both the disruptive potential of technology, including threats and opportunities, and the sustainability agenda needs [27,33]. Moreover, the disruptive nature of technological advances accentuates the need for new governance mechanisms (regulatory, normative, infrastructural) to ensure the avoidance of the potential challenges, including increasing citizens’ acceptance of and reliance on such measures [27,32,33]. 

Based on these challenges, the research objective of this article was to gather, understand and compare perceptions of citizens of five European countries towards a scenario of transition to healthier, more equitable and sustainable lifestyles in Europe by 2040. Scenarios of the future can be useful tools to gain insights for developing policies to either get to those futures or to adapt strategies for these futures. Narrative storylines around different indicators can help citizens and policy makers to engage with different possible futures and to consider their preferences for future developments in a number of fields, including health [34,35,36,37]. In this scenario, titled “My life between realities,” technological solutions are a driving factor of this transition; hence, the focus of this paper on this scenario. The “My life between realities” scenario has been developed in conjunction with three other scenarios, in a previous study carried within the INHERIT project by Guillen-Hanson et al. [38]. (The EU-funded project INHERIT aims to examine the possibilities of changes in lifestyles and behaviors to encourage the transition towards healthier, more equitable and sustainable societies (see Staatsen et al. [18] and www.inherit.eu).) Each scenario suggests different pathways for reaching healthier, more equitable and sustainable European societies by 2040. The development of the scenarios preceded the citizen consultations which are discussed in this paper. Nonetheless, since the scenario(s) were central to the discussions, the research team deemed it necessary to give readers a better contextual overview of this activity preceding the focus groups discussions. Thus, the three other scenarios discussed and the scenario planning process are briefly introduced in this paper. 

The paper is structured as follows. First, we present the materials and methods, including discussion of the development of scenarios and the design of focus groups to discuss these. We then present the results, before a discussion of the implications of these for policy. Finally, we present conclusions and areas for future research. 

## 2. Materials and Methods 

### 2.1. Scenario Planning 

Scenario building or planning is a foresight method for anticipating future developments [39]. It supports medium to long term strategic planning and decision-making processes while accounting for future uncertainties and complexities. The latter are seen as driving forces, trends, values and/or external shocks throughout different areas such as social, economic, technological, environmental and/or political systems [40]. Scenario planning does not involve predicting the future, but rather illustrating potential futures. Usually, two to four scenarios are created depicting methodologically researched and developed narratives about what the future could potentially look like [41,42]. The open and flexible nature of the method allows for its diverse utilization in terms of context and purpose. Moreover, this also enables the customization of the methodological process depending on the needs and time [41,42,43,44]. For the development of the scenarios which preceded the citizen consultations through focus groups, the scenario planning process of European Foresight Platform [42] has been adopted and aligned (see Figure 1). 

#### 2.1.1. The Setting of the Scenarios 

The first step of a scenario planning exercise is to define its purpose or aim and the time horizon [42]. The purpose of this study was to find out what healthier, more equitable and sustainable lifestyles in Europe could look like by 2040. This specific period was selected to represent a medium-term future that captures and goes beyond the implementation of Agenda 2030 for Sustainable Development. This question was explored against three main lifestyles areas; namely, living (green spaces and energy efficient housing), moving (active mobility) and consumption (sustainable food and nutrition). 

#### 2.1.2. Identifying and Analyzing Drivers 

The second step of a scenario planning exercise is to identify and analyze factors (trends and drivers) that are relevant for the context and purpose of the research [42]. In this case, the focus was on factors influencing Europe’s health, environmental state and equity degree. For the research study, a combination of STEEPLE (social, technological, environmental, economic, political, legal and ethical change) and the horizon scanning approach was used. Both are foresight tools which in combination allow for the exploration and identification of early changes important for future developments, including threats and opportunities. The scanning and identification of trends were conducted through desktop research, and a first round of evaluation was conducted by researchers in the field. The output of this exercise was a compilation of several impactful trends in Europe towards 2040 [45].

#### 2.1.3. Ranking by Perceived Impact and Expected Uncertainties 

Following the first round of evaluations, in a qualitative exercise, the consolidated trends were evaluated in another round by pan-European experts coming from the academia, civil society, policy and business. Trends were evaluated based on impact and uncertainty. The high impact and low uncertainty evaluated trends were used to shape the narratives of the scenarios (e.g., ageing population and energy transition), whilst the high impact and high uncertainty trends determined the scenario extremes (e.g., use of virtual/augmented realities and urbanization). 

#### 2.1.4. Setting the Parameters 

To further develop the scenarios, it is necessary to define the parameters or the dimensions based on which scenarios will be constructed [42]. Based on the previous exercise and the research needs, two dimensions, namely, social and sectoral, were identified. These reflect the most influential drivers and the diverse organizational (governance) approaches within the EU as a construct (see Figure 2). The dynamics of the social dimensions can range between individualistic and collectivist. The sectoral dimension reflects the type of governance in a country with dynamics ranging between private (led by market actors) and public (led by governments) governance forms.

#### 2.1.5. Elaborating the Scenarios

In a next step, the narratives of the four scenarios based on the output of the previous steps were developed. These narratives illustrate lifestyles occurrences throughout the three areas (living, moving and consuming) and describe the everyday life of one particular fictional citizen living in each scenario. The names of the scenarios reflect their attributes. The four scenarios were: (1) my life between realities (private sector, individualistic social processes); (2) less is more to me (public sector, individualistic social processes); (3) one for all, all for one (public sector, strong collectivism); (4) our circular community (private sector, strong collectivism).

It is important to reinforce that the scenarios depict positive visions of healthier, more equitable and sustainable European societies while not focusing on any particular country or region in Europe. The main characteristics of the scenarios are described in Figure 3. 

#### 2.1.6. Implications of the Scenarios 

As a last step, the research team focused on understanding the strategic implications of the scenario narratives, including the development of strategies and action plans for future policies. As such, the scenario’s challenges and opportunities for health, equity and sustainability in Europe were elaborated. 

#### 2.1.7. My Life between Realities

In this paper, we focus on the “My life between realities” scenario which is characterized by digitalization, including virtual and augmented realities, artificial intelligence and big data, inter-connectivity and personalization. The social dynamics in this scenario are individualistic, that is, the individual (with its needs and aspirations) is at the center of the processes, while, on the sectoral dimension, businesses drive the governance approaches complemented and supported by governmental interventions. The government’s role is limited to ensuring ethical, egalitarian and uninterrupted social and economic operations. Technology and related advancements, such as big data, artificial intelligence and virtual (augmented) reality are used to increase (resource) efficiency and operational performance and to provide citizens with qualitative and personalized (need based and preventive) products and services for better and healthier living. 

##### Living: Green Spaces

In “My life between realities”, a small share of green spaces is virtual, enabled and sponsored by companies. Companies also provide the VR glasses that citizens can use to experience virtual green spaces and have the possibility to relax. In this way, green spaces are accessible to citizens who do not have easy access to parks or forests. Physical green spaces are also available for citizens, provided jointly by governments and companies; however, citizens visit these only sporadically. 

##### Living: Energy Efficient Housing 

Homes in this scenario are smart, leading to optimized energy consumption that is sourced by large scale renewable energy installations. All energy-using devices in the household are virtually connected and the inhabitants’ behaviors are monitored to offer personalized, need-based and efficient solutions. Large companies oversee the market and provide offers in all price segments. 

##### Moving: Active Mobility 

A highly connected, electrified and autonomous transport system characterizes the mobility attributes of this scenario. Interconnected public transport is the dominating mobility means in this scenario, complemented by self-driving cars for the areas where public transport does not reach (i.e., rural) as well as biking and walking. Citizens are financially incentivized to use public transport, bike and walk by companies and health insurance schemes, while the same means are used to discourage motorized transport. 

##### Consumption: Food and Beverages 

In this scenario, citizens are able to follow personalized diets based on their health conditions and needs. Meat is produced in laboratories by scientists using cells from actual animals; thus, ensuring animal welfare. Product value chains are transparent and accessible to consumers. To offer personalized products, large companies have an increasing knowledge about consumers’ food preferences and health needs.

### 2.2. Focus Groups

To explore citizens’ perceptions, the research team conducted focus groups in five European countries: the Czech Republic, Germany, North Macedonia, Spain and the United Kingdom. Focus groups were deemed an appropriate way to elicit preferences and opinions about the different scenarios. Focus groups as a technique, ensure both the necessary scientific approach for a research study and the collection of sufficient information to analyze complex topics that do not necessarily qualify for quantitative analysis [46]. The interactive trait of focus groups enables the collection of diverse paradigms and perspectives, while allowing participants to reflect, compare and be introduced to perceptions other than theirs [47,48,49]. As such, in focus groups, participants are able to complement and expand their reflections, leading to more comprehensive feedback on the research topic [46,50]. Moreover, focus groups enable the gathering of information in a relatively short amount of time and with limited financial means [49,51]. 

Three focus groups, between six and eight participants each, were used in each aforementioned country, with 15 focus groups and 118 participants in total. The countries were chosen to give a wide geographic coverage across Europe and to ensure cultural and socioeconomic diversity from across Europe. 

To ensure a balanced and diverse discussion, the research team aimed for a heterogenous sample of participants in terms of sociodemographic characteristics (i.e., gender, age and income) which have been summarized in the Table 1 below. Moreover, in this study, the research team aimed at collecting insights from non-experts. Thus, several exclusion criteria pertaining to participants’ professions were applied. Accordingly, citizens working in the food production, urban planning, car manufacturing and/or medical areas were excluded from participating.

The interactive trait of focus groups and the presence of other people can lead to socially biased opinions or answers. Accordingly, the fear of disapproval, and conversely, the desire of approval by other participants, could lead to normative answers and discussions [49]. Depending on the complexity of a topic, semantic issues might arise also, leading to unreliable feedback and results of doubtful quality [49]. Moreover, there is the possibility of focus group participants focusing their discussion and contributions on some aspects of the research topic, while avoiding others, leading to an incomplete feedback, and thus, analysis of the topic [49]. Such limitations were carefully considered during the design stage of the focus groups. Accordingly, the focus groups in all five countries followed and were based on a structure pre-defined by the research team. Focus groups were moderated using a facilitator guideline and participants’ exchanges were driven and supported by visual tools such as videos: (a) one introductory and summarizing video for all scenarios; (b) four videos (the videos are available at https://www.inherit.eu/future-scenarios/) showing each of the four future scenarios; and (c) print-outs describing in more detail, the lifestyle areas of each scenario. Participants discussed their perceptions after each video was shown to them. The focus groups had semi-structured group discussions addressing each lifestyle area depicted in the scenarios. The moderator asked open-ended questions to entice and engage all participants in the discussion based on a topic and to ensure the exploration of more in-depth themes and responses. The moderators were also instructed to observe and clarify any unclarities the participants might have had and to ensure that all thematical aspects of the research were proportionally covered. To enable free and uninterrupted discussions, the focus groups were conducted in the countries’ national languages. All the materials used in the focus groups were carefully translated (from English to national languages) to avoid content misinterpretation. The video audio remained in English; however, with subtitles in national languages. 

The data from all five countries were digitally transcribed and translated to English by the focus group conductors. The data were analyzed by applying a qualitative content analysis method, based on a constant comparison analysis rationale [52]. By using this methodology, the research team was able to develop a theory more or less inductively, by categorizing, coding and delineating categories, and connecting the data which emerged from the focus groups. Accordingly, constant comparison went hand in hand with the theoretical sampling principle. This enabled the research team to answer questions that arose from the analysis and to reflect on the data. Such questions concern interpretations of phenomena or finding relations between categories. In this study the data categorization and coding were performed through a qualitative analytical software MAXQDA Analytics Pro. Its application enables the development of datasets building upon the INHERIT Common Analytical Framework (CAF) [18], which includes the behavior change wheel (BCW) model [53]. The CAF, on basis of BCW, in a holistic and systemic manner, illustrates the complex and interactive, causal relationships, including implications between our physical surroundings, health, health equity and environmental sustainability, and related human behaviors [18]. According to the BCW, a behavior is the interplay between several internal and external factors clustered overall as capability (human physical and psychological skills/abilities), motivation (human cognitive processes, inducing one’s behavior) and opportunity (external factors dictating the implementation of a behavior) [18,53]. As such, these models can be utilized for assessing the efficacies of policies and other solutions that aim at increasing the sustainability of our living patterns, and support the leveraging of other opportunities of change that lead to such progress [18]. The dataset of codes utilized for the qualitative content analysis, in this study, reflected these variables, and were complemented by the key units of analysis; namely, the INHERIT scenarios and related lifestyles areas. Furthermore, to better understand phenomena and boundaries, the constant comparison analysis also included other additional variables that emerged; namely, participants’ perceived fears, risks and challenges; policy recommendations; and participants’ preference towards the actors driving the scenarios. The final codebook can be found in Table A1 in Appendix A. 

## 3. Results

The analysis of the data collected revealed the citizens’ attitudes towards the “My life between realities” scenario were to a large degree diametrically opposed, with disapproving tendencies. These critical tendencies came as a result of the scenario’s high reference to and reliance on technological developments. Virtual reality, big data and increased integration of automated processes in everyday living occurrences, as well as monitoring of citizens’ behaviors for offering personalized products and services were deemed quite controversial by citizens in all five countries. The general backbone rationale behind these attitudes involved the fear of unknown processes, the loss of experiential authenticity and diminishment of autonomy, violation of privacy and the potential for increased social detachment or isolation. Moreover, participants were rather doubtful about the increased role of private actors (i.e., companies) in overseeing and driving some of the main aspects of this scenario.


**Quote:**
*“I don’t know if it is possible or not [the scenario “My life between realities”], but I hope it doesn’t happen. It doesn’t provide you with what it should; it dehumanizes”*
*(Spain, 32, female, middle income).*

Attitudes of this nature were present throughout the entire discussions related to the scenario; however, (some) participants were able to identify and recognize opportunities, also. Accordingly, augmentation of lifestyles with technological measures, according to focus group participants, would overall lead to increased operational efficiency; thus, leading to resource, financial and time saving opportunities. The latter could be used for additional activities that would increase citizens’ quality of life, well-being and satisfaction. Moreover, these technological advancements were seen as contributing highly towards improving and maintaining steady conditions of good health. Participants were able to think and bring forward beneficial and opportunistic aspects, especially during the discussion rounds focused on the scenario’s particularities (i.e., elaboration of occurrences throughout the four lifestyle areas). 


**Quote:**
*“The use of technology at home is nice; that it can help you, take different worries off you, the need to think. So, you have more time for yourself, for your family.”*
*(Czech Republic, 37, male, low income).*

### 3.1. Green Spaces 

Cumulatively, in all five countries, participants expressed attitudes of dissatisfaction towards the utilization of virtual green spaces as complementary or substitutes for physical green spaces. Diminishment and loss of experiential authenticity, including preference for the physical nature, poor stimulation of senses and inability to perform activities as one would do outdoors, were the main drivers opposing this development. 


**Quote:**
*“As the most negative, I have marked the first scenario [“My life between realities”], since in my opinion, virtual reality cannot be compared with the real one. It is not the same.”*
*(North Macedonia, 26, male, middle income).*

Moreover, the role of companies in enabling both the virtual and physical green spaces in this scenario, thereby increasing their control throughout all areas, complemented the reasons contributing towards the negative perception among some of the British participants. These critical attitudes, for the developments in this area, persisted throughout all discussion rounds with limited elaboration on opportunities. Nonetheless, some of the British and German participants recognized the substitute character and beneficial contribution of virtual greens spaces for the elderly and disabled people who may be limited in their access to physical green spaces. 


**Quote:**
*“It would give disabled people that chance to do things; that, probably, is its only benefit that I could think of.”*
*(UK, 62, female, high income).*


**Quote:**
*“I could imagine it very well for people in an old people’s home who can’t get out anymore.”*
*(Germany, 58, female, middle income).*

### 3.2. Energy Efficient Housing 

The technological augmentation of homes to support optimized energy consumption was subject to diverse opinions by the focus group participants in each five countries. Participants in the Czech Republic and North Macedonia expressed mainly positive attitudes. These positive attitudes centered around the potential of technology to increase efficient consumption, and thus lead to financial benefits for household residents. Moreover, in the North Macedonian focus groups it was indicated that technological solutions (i.e., smart homes) may be more effective in this aim than any other efforts on a consumer/household level. German, Spanish and British participants recognized the ability of smart homes to increase the residents’ convenience, and furthermore, British participants discussed the potential of these solutions to increase residents’ ability to understand and monitor their energy consumption. 

**Quote**: *“[...] I think it’s good if a lot of things are handled automatically in the house for me. You come home and it knows it’s dark now, it’s winter and the light switch on. But you shouldn’t give up control. I should program it beforehand so that you don’t have to pass this data on, and if I’m at home for the weekend then it should be like this when I’m away to start the washing machine. I think it’s great; why not? It makes life easier.”**(Germany, 40, male, low income).*

However, participants also found the potential of increased monitoring of and data collection on individual behavior worrisome. Thus, deeming smart homes a good but unnecessary addition. This attitude was furthermore enforced, among the British and German participants, due to their distrust towards companies and beliefs that companies have the tendency to violate one’s privacy, which were described as responsible for overseeing these developments within this area. Spanish participants shared the same concerns about companies’ roles, with the additional concern about a perceived tendency for unilateral price setting policies. One Czech participant expressed their worries about technological failures and related implications on one’s everyday live. Worries about the unequal distribution and access of these services by all citizen groups due to presumed high costs were expressed by Czech, German, Spanish and British participants also.


**Quote:**
*“Well, big companies created a set of proposals, so it is clear that they will cost and it is clear that big corporations will have them hugely overpriced, so this is what I don’t like, that big corporations would be in charge because it would be the same as today. It would be inaccessible for 99% of people...” *
*(the Czech Republic, 27, male, low income).*

### 3.3. Mobility 

The “My life between realities” mobility attributes, such as interconnectivity and efficient systems with public transport as the main mode of motorized transport, received approving opinions by the participants of all five countries. Increased efficiency and opportunities to save time complemented by increased convenience and mobility without limitations were some of the most frequently mentioned factors. Nonetheless, influenced by current operational patterns, namely, low reliability and poor sanitary conditions, Czech, British and German participants brought forward doubts about impactful future increases in the utilization of public transport for mobility purposes. The scenario’s reliance on financial means to support and increase the share of walking and biking as a means of mobility was positively confirmed by the North Macedonian and Spanish participants and not discussed by the others. North Macedonian participants recognized this scenario’s potential by using disincentivizing financial measures to make cars unwanted; thus, contributing to better environmental and human health conditions. 


**Quote:**
*“Because measures like incentivizing public transport prices or facilitating connections with areas further away are also great measures for mobility.”*
*(Spain, 22, male, middle income).*

The autonomous driving cars central to this scenario were appraised positively by the majority of Czech participants with convenience and efficiency as supporting factors. However, this view was not shared by German and British participants. The German participants based their disapproving stance on the current low diffusion and implementation levels of self-driving cars, as well as the presumed high risk of technological failures. In addition to these, British participants had concerns about the risk of software hacking and related implications. Discussions on this particular topic were lacking in the North Macedonian and Spanish focus groups. 


**Quote:**
*“And what if the car breaks down on the road, god forbid, and I don’t have a computer to do an analysis of the problem.”*
*(the Czech Republic, 57, male, high income).*

### 3.4. Food 

The drive and aspiration for maintaining good and/or improving health conditions and lifestyles led to focus group participants in all five countries expressing positive attitudes towards personalized diets and nutrition, one of the key elements of the consumption lifestyle area in the “My life between realities” scenario. Additionally, Czech and British participants thought that augmenting this lifestyle area with technological advancements could lead to increased convenience. Czech participants framed this as support for citizens in terms of better managing their time in view of demanding lifestyles also and advantageous for people in elderly age. Similarly, some British participants thought these technologies could ease people’s management of their various activities, as well as support those who lack dietary and food handling knowledge. 


**Quote:**
*“Well I like the idea of the first one because it says you can personalize your nutrients so then your food, like your health conditions. I think that would give people lot better life quality as well, less illnesses; less side effects from illnesses would give people a better quality of life.”*
*(UK, 58, male, low income).*

Nonetheless, participants in the German focus groups expressed strong concerns about violation of personal privacy due to the monitoring of behavior necessary for offering personalized dietary solutions. In addition, some German and Czech participants thought these developments could lead to food consumption patterns of a functional nature, while dismissing elements related to pleasure-based eating. Moreover, the dominant role of companies in overseeing these processes was opposed by the Germans as well as the Czech and Spanish participants. A controversial element of this lifestyle area in this scenario was the laboratory produced meat, with Czech participants expressing doubtful attitudes, while fear of negative health impacts driving the Macedonian and Spanish focus group participants’ negative attitudes towards the idea. 


**Quote:**
*“I did not like the fact that meat was produced in laboratories. It may need to be examined whether it has negative effects on the human organism.”*
*(North Macedonia, 36, female, high income).*

Many British participants opposed the idea due to finding this method of meat production overly artificial. However, at the same time, the method’s potential to promote animal welfare was recognized by some British and Spanish participants. Higher value chain transparency degree was discussed only briefly by German and Spanish participants and not at all in the other countries. Some of the German participants found the idea irrelevant for their values, while Spanish participants expressed approving attitudes. 

## 4. Discussion

The research objective of this study was to gather, understand and compare perceptions of citizens from five European countries towards one of four future scenarios depicting a vision of a healthier, more equitable and sustainable Europe by 2040. This scenario called “My life between realities” predominantly relies on and suggests technological measures in reaching this vision of the future of Europe. Accordingly, the discussions with 118 participants of 15 focus groups in total, conducted in five European countries, show that citizens hold dichotomous attitudes or opinions with disapproving tendencies towards the scenario “My life between realities” and the increasing adoption of technological measures in one’s day-to-day living. The diverging opinions were conditioned by the different occurrences throughout the four lifestyle areas; namely, green spaces, energy efficient housing, active mobility and food. For example, in the context of autonomous (self-driving) cars and laboratory produced meat, citizens expressed concerns about technologies’ potential negative impact on health and safety. Nonetheless, the opportunity and motivation of maintaining and/or improving their health conditions and well-being led to citizens expressing positive opinions towards personalized diets and nutrition. Similar attitudes, as elaborated in the results section, persisted, in various degrees, throughout all lifestyle areas. Thus, the diversity of attitudes and opinions deems necessary, reciprocally responsive policy approaches to match citizens’ needs and to address concerns systematically.

Overall, citizens’ concerns can be broadly clustered (a) based on their perceptions about technologies’ potential negative impact on one’s privacy or (b) individual autonomy; (c) overarching and reiterating concerns about the dominant role of private sector actors in leading these developments; (d) perceptions about technologies’ negative impacts on health and well-being as well as on (e) social interactions/relationships. The persistence of these negative attitudes and opinions might seriously inhibit the diffusion of technological advances in the domains of green spaces, energy efficiency, mobility and food consumption. Technological advances hold great potential for rectifying some of the negative impacts of our societies’ unsustainable production and consumption patterns. Accordingly, there is a need for policy action frameworks, driven by various stakeholders and on various levels, that aim at changing public perceptions or attitudes and improving trust towards technological developments and related operations.

Citizens’ in all five countries, driven by their preference for maintaining their privacy, reiterated most frequently, their concern about data collection, analysis and handling for the purpose of offering personalized products and services and optimizing resource efficiency. Accordingly, increasing citizens’ acceptance of these solutions would require the adoption of (regulatory) policies that aim at ensuring the secure management and ethical governance of their data as well as establishing and enforcing grievance mechanisms to ensure compliance. Moreover, through various communicative means, citizens need to become aware of the existence of such approaches and mechanisms to maintain personal privacy. 

The subjective perception of inclusion could also improve the acceptance of technological change that might otherwise be diminished due to concerns about the reduction of individual autonomy. The focus groups results indicate that an active engagement of the public audience (i.e., citizens) should be implemented in all phases of the product/service development, also, thereby, increasing their understanding about these products/services, giving them the opportunity to familiarize with otherwise unknown processes and capture their needs and requirements in the very early stages. The opportunity of overruling automatic functionalities of technologies in favor of manual operations should be available to citizens throughout the entire duration of the use-phase. Such policies would enable and ensure citizens’ autonomy.

In other multiple instances, citizens expressed distrust towards the scenarios’ occurrences due to the prominent role of private actors in driving those. Following this, for maximal diffusion, technological devices, solutions and digital systems should be developed and tested in conjunction with and endorsed by other experts and stakeholders as well (i.e., adopting interdisciplinary and multi-stakeholder approaches throughout all product/services development stages). These efforts, as elaborated previously, could potentially be stronger if citizens themselves were to be engaged in this process. The endorsement and validation of such products by independent experts, in turn, would lead to reduced distrust in relation to the products’ potential negative impacts on health and safety. 

Undoubtedly, technological or digital products and services are challenging the conventional norms of social interaction. However, such developments, as shown throughout the years, have the potential for connecting people without much geographical limitations, and/or, as shown by Ballantyne et al. [54], reducing feeling of loneliness, especially among elderly people. Nonetheless, to address citizens’ legitimate fears that digitalization might lead to social isolation and/or detachment, developers and innovators could integrate more interactive features when designing new digital products and services.

Similarly, the perceived benefits or opportunities of citizens in this scenario could be clustered on basis of technologies’ positive impact on (a) operational and resources efficiency; (b) financial and time saving opportunities; (c) citizens’ health, well-being and life quality; and (d) convenience. The identification of these aspects or opportunities during the discussions, led to citizens expressing higher acceptability and willingness to adopt some of the measures described throughout the lifestyle areas of the scenario. While previous research indicated support for positive financial measures (subsidies) and opposition towards their negative counterparts (taxes), during the workshop, positive views of charges was recorded. Accordingly, centering the communication and placing the technological products and services in the context of these factors and making the latter easily visible could improve citizens’ acceptance of the former. 

This exploratory examination of citizens’ perceptions is based on qualitative research that is more focused on the meanings, interpretations, and explanations of people rather than on generalizability of the results [55]. Due to small participant sample and nature of this study, results of this paper cannot be generalized to the populations from which our samples were drawn and/or to populations of other (global) regions. However, our study aimed at fulfilling the criteria of quality for qualitative research; that is, trustworthiness, credibility, authenticity and plausibility [56]. In order to provide robust qualitative findings, we paid special attention to each of the research steps: (1) research design; (2) data collection; (3) analysis; and (4) reporting of results. Further, the findings of this article can be supported by the conclusions of a quantitative survey, with 12,288 respondents in five European countries (the Czech Republic, Latvia, Portugal, Spain and the United Kingdom) that elicited citizens’ opinions to the same four scenarios introduced in this article, including here “My life between realities” [57]. Accordingly, the scenario “My life between realities” generated the same diametrically opposed perceptions, especially in relation to the green spaces and food lifestyle areas. Only 4% to 6% of respondents supported the vision of virtual green spaces, while only a minority of respondents (9% to 12%) were in favor of the vision of laboratory produced meat and personalized diets. A larger share (20% to 17 %) selected this scenario in case of energy efficiency. On the other hand, this scenario was appreciated as a good vision for active mobility [57].

## 5. Conclusions

Citizens of five European countries involved in focus groups, in general, expressed dichotomous attitudes towards the increasing role of technology in one’s living patterns. Concerns about technology’s potential negative impact on privacy, autonomy, human health and safety and social cohesion, drove citizens’ doubtful attitudes. Additionally, the increasing role of the private sector over the public one in these lifestyle dynamics accentuated the concerns. However, the contribution of technology in operational efficiency leading to savings in financial and time resources, its potential to maintain or improve one’s health and well-being and to increase one’s convenience and support in completing daily activities, were some of the opportunities recognized. Lay knowledge of risks might be more intuitive and less formal and precise compared to expert perceptions; however, even then, it is important for policy makers and other parties to be aware and consider these in their operations. Changing these perceptions and increasing citizens’ acceptance of technological solutions is important for leveraging the latter’s potential in advancing sustainable production and consumption patterns. Accordingly, the character of concern should match the type of response. The policy implications elaborated in this article could support the development of initial action frameworks necessary in times of such socioeconomic transformations induced by technological advances.

This research study contributes to understanding better citizens’ perceptions of technological developments as a potential measure for reaching more sustainable development. Nonetheless, due to the foresight character of scenario planning or building, the developments are rather theoretical and hypothetical and so are the citizens’ perceptions and perception formations. Accordingly, further research is needed regarding citizens’ perceptions and attitudes, including potential adoption rate, towards technological measures they are able to experience and utilize. Moreover, as already highlighted, this study presents the insights from a diverse range of participants in five European countries. This implies that the views recorded correspond to a limited series of backgrounds and to a somehow specific geographic context, which can be seen as a limitation when trying to extrapolate conclusions to populations of the same or different global regions. It must also be noted that this is an exploratory examination of citizens’ perceptions, and therefore, results should be treated with caution. Further research is needed to validate these results in different contexts and in grander population samples. Such knowledge would complement and contribute to better and more adequate responses for increasing citizens’ acceptability of technological solutions, which in turn would bring us closer to reaching the vision of healthier, more equitable and sustainable European and global societies. 

## Figures and Tables

**Figure 1 ijerph-17-00231-f001:**
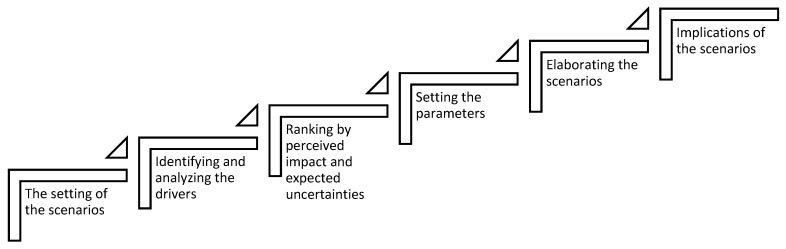
The scenario building process (adapted from the European Foresight Platform (EFP), n.d. [42]).

**Figure 2 ijerph-17-00231-f002:**
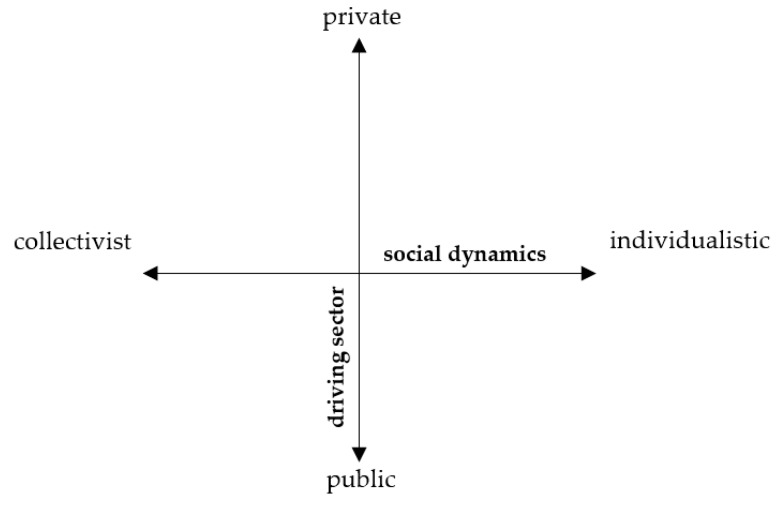
Dimensions of the four future scenarios.

**Figure 3 ijerph-17-00231-f003:**
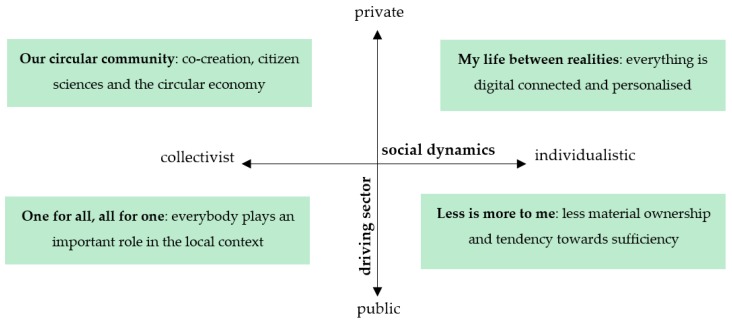
Overview of the four future scenarios.

**Table 1 ijerph-17-00231-t001:** Summary of participant’s key socio-demographic characteristics.

	Czech Republic	Germany	North Macedonia	Spain	United Kingdom
Female	12	12	12	13	10
Male	12	12	11	11	13
High income	10	8	8	4	6
Middle income	3	8	7	14	8
Low income	9	8	8	6	9
N.A.	2				
Senior (65–100)	0	3	1	2	2
Adult (30–65)	21	16	18	13	17
Youth (18–30)	3	5	4	9	4
Total per country	24	24	23	24	23
Total overall	118				

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
