# Peer review of "A Technological Scenario for a Healthier, More Equitable and Sustainable Europe in 2040: Citizen Perceptions and Policy Implications"

_ijerph, 2019, doi:10.3390/ijerph17010231_

Round 1
Reviewer 1 Report
Dear Authors, the paper is very interesting but needs some minor revision:
1) The Abstract - it is lack of the main goal of the study - this needs to be added
2) The methodology part - what are the advantages and disadvantages of using this method
3) The Conclusion - limitations of the research and future studies needed in this field needs to be more accurate
4) English - I encourage You to use an native speaker to check the grammar etc. of the paper
Good luck with the revisions.
Author Response
Dear Reviewer,
Thank you very much for your feedback! We are glad to hear you found the paper interesting!
As deemed necessary, please find below some responses to your comments in respect to the paper. Please note, I abstained from repeating your comments in the responses.
1). The previous abstract contained the objective, but I can see how it was not very straightforward. Thus, I changed slightly the first sentence to make the objective clearer. The first five and a half sentences of the abstract are dedicated to the objective of the study (lines 22-27).
2). Some advantages and disadvantages of the focus group methods are now included in the revised version (section 2.2). We limited this input to the focus group method only, since the latter is the main method of this particular study.
3). The paper has been proofread by a native English speaker.
Thank you and best regards!
Reviewer 2 Report
This paper investigated the citizen perceptions and policy implications on a technology-driven scenario under an EU project, INHERIT.
Overall, the research is well designed.
The findings from focus group discussion encompassing 5 countries and 118 people may be an interest to some readers.
However, before publication, here are some of my concerns:
1. The scope of study is slightly narrow, focusing only on the technological scenario. In this case, more in-depth discussion on the findings and policy implication are expected. I found the current content too thin and must be enriched (I will elaborate in later points).
2. The introduction is very long and hard to read. Please consider to use a subtitle to break them into two parts: first, the background of European development; and later, the background for technological scenario (digital revolution etc.)
3. I found this sentence slightly confusing. "It differs in this respect from three other scenarios developed by Guillen-Hanson et al. [38] which are briefly introduced in this paper..." It seems to suggest that the three other scenario were developed by GH et al. but the technological scenario was designed originally by the authors. Please clarify and consider to paraphrase.
4. The section 2.1 scenario planning is very long and has too much similarity to the reference Guillen-Hanson et al. [38]. Is this really needed? Also, it must be clearly stated upfront that the scenario planning was originally developed in previous study.
5. In Figure 2, the title for x-axis and y-axis are located in the quadrants. Perhaps the location overlapping with the axis is more clear (optional), like the graph shown in Guillen-Hanson et al. [38].
6. The subsection numbering is wrong. Missing subsection 2.1.7.
7. The original contribution of this paper is in the focus group discussion, in my opinion (the rest are partially published somewhere else?). Therefore, a clearer description of: how the participants were selected, why these countries, how representative with this small number of participants, or what limitation we must pay attention to when interpreting the result due to the sample size etc.
8. Please consider add an figure that shows one or two photos of the focus group discussion activity.
9. The analysis using MAXQDA was described briefly with references. But we cannot understand how exactly the analysis is performed. Please clarify.
10. It seems that the results from MAXQDA were not specifically presented in the result section.
11. In addition to item 10, the results should be elaborated. In each subsection, the authors occasionally described some discrepancies between participants from different countries. Are you implying the differences were due to factors of each country? If so, a table summarizing the differences among responses from different focal groups on key categories will improve the readability of the section.
12. The title said, "... citizen perception and policy implications." I presume the discussion section is dedicated to discuss the policy implications since this is not found in the results. But this is not clear in the discussion. Please consider a more straightforward writing.
I would recommend the paper if the above points can be addressed sufficiently.
Author Response
Dear Reviewer,
Thank you very much for your feedback to this paper!
As deemed necessary, please find below some responses to your comments in respect to the paper. Please note, I abstained from repeating your comments in the responses.
1 and 12) Further content and elaboration in the discussion section has been included. Also, the language and text has been revised to reflect the policy implications more prominently.
2) I broke down the introduction in two sections as per your recommendation.
3 and 4). I have rephrased the sentence to make sure the reader understands that all scenarios have been developed by Guillen-Hanson et al. (2018) in a previous study. I can fully understand your point about the section on the scenario planning being too long, however, we felt a proper elaboration on this is necessary to ensure the reader understands the scientific approach behind such process and understand a bit more the context preceding the focus groups methodology and this study.
5). Really good suggestions. I attempted to follow it, however, design wise it did not look so appealing in this case, due to the color limitations.
6). The numbering is fixed in this version.
7). The paper has been revised according to your suggestions: reference to participants and country selection criteria have been included; also, we have elaborated on the representativeness (or robustness) of the study given the small number of participants; related to the latter, we also have elaborated on the limitations of the study and future research recommendations (throughout section 2.2, section 4-discussion and 5-conclusion).
8). Unfortunately, we are not able to include photos of the focus group activity, since we are bound to the anonymity of the group (data protection requirements we set to ourselves). Any photo where the participants are not detectable potentially will not look good (design wise).
9). Further elaboration on how the analysis has been conducted has been included.
10). MAXQDA was only the software used for the results analysis. The result section has been developed on basis of this analysis. Since MAXQDA was only the software, we deemed it better if don’t refer to it specifically in the results section.
11) Due to the small participant sample, we do not intend to claim that the differences between countries occur due to country specific factors. Such differences come across only because we try to report on the analyzed attitudes in the various countries, however, without aiming to cross-compare the countries in the strict sense. However, we did try to bring forward some of the most common reasons for the positive or negative attitudes towards the scenario, which we tried to make more prominent in the discussion section of the revised article.
Thank you and best regards!
Reviewer 3 Report
Dear Authors,
First of all, well done!
Please consider my comment below carefully:
In (2.2 Focus Groups), you have to explain how this small size data pattern (118 in total) can lead to a robust prediction? Giving that you are considering four major fields: Green Space s Energy-Efficient Housing Mobility FoodHave you found a clear difference between Senior and Young people responses?
Author Response
Dear Reviewer,
Thank you very much for your feedback! We are glad to hear you found our work well-done!
As deemed necessary, please find below some responses to your comments in respect to the paper. Please note, I abstained from repeating your comments in the responses.
1). In the revised article we have elaborated on how the small size sample can lead to a more robust results/predictions (section 4-discussion).
2). Unfortunately, such comparisons were not part of the research design per se. From the demographic characteristics one can observe discrepancies between the participation rate of senior and young members. Thus, any efforts of including such observation would not lead to any scientifically based findings.
Thank you and best regards!
Round 2
Reviewer 2 Report
Nice work with the improvement. I recommend to accept after addressing the error noted below.
Please check line 593 for some technical citation error.